

# Exploring the potential of deep-blue autofluorescence for monitoring amyloid fibril formation and dissociation

Mantas Ziaunys, Tomas Sneideris and Vytautas Smirnovas

Vilnius University, Life Sciences Center, Institute of Biotechnology, Vilnius, Lithuania

## ABSTRACT

Protein aggregation into amyloid fibrils has been linked to multiple neurodegenerative disorders. Determining the kinetics of fibril formation, as well as their structural stability are important for the mechanistic understanding of amyloid aggregation. Tracking both fibril association and dissociation is usually performed by measuring light scattering of the solution or fluorescence of amyloid specific dyes, such as thioflavin-T. A possible addition to these methods is the recently discovered deep-blue autofluorescence (dbAF), which is linked to amyloid formation. In this work we explore the potential of this phenomenon to monitor amyloid fibril formation and dissociation, as well as show its possible relation to fibril size rather than amyloid structure.

## INTRODUCTION

Amyloid fibrils are highly structured protein aggregates (*Fitzpatrick et al., 2013*) associated with several neurodegenerative disorders, such as Alzheimer's, Parkinson's and prion diseases (*Chiti & Dobson, 2006*; *Knowles, Vendruscolo & Dobson, 2014*; *Chiti & Dobson, 2017*). Determining the rates of protein self-association into amyloid aggregates is important for understanding possible aggregation pathways (*Meisl et al., 2017*) as well as for identification of potential inhibitory substances (*Porat, Abramowitz & Gazit, 2006*; *Arosio et al., 2014*). Furthermore, it is well known that the tightly packed fibrils are stable in denaturant solutions up to a certain denaturant concentration (*Vernaglia, Huang & Clark, 2004*). This factor can be exploited to distinguish differently structured aggregates (*Sneideris, Milto & Smirnovas, 2015*).

Typically, association/dissociation of amyloid fibrils is monitored using an amyloid-specific dye thioflavin-T (ThT) assay (*Xue et al., 2017*). ThT molecules bind to the $\beta$-structure grooves on the surface of fibrils and attain a stabilized conformation, which greatly enhances their fluorescence potential (*Biancalana & Koide, 2010*). The rationale behind this method is that the fluorescence emission intensity is proportional to the amount of ThT molecules bound to fibrils and as the number of aggregates changes, so does the amount of ThT molecules with enhanced fluorescence, allowing for an easy way to track their concentration. While this approach is simple and widely used (*Narimoto et al., 2004*; *Cobb et al., 2014*; *Sun et al., 2008*; *Biancalana & Koide, 2010*), there are multiple

Corresponding author
Vytautas Smirnovas,
vytautas.smirnovas@bti.vu.lt,
vytautas@smirnovas.info

factors that can influence the assay's outcome. First, the denaturant can influence ThT affinity to fibrils and its fluorescence properties (*Kuznetsova et al., 2012*; *Voropai et al., 2003*). Moreover, the dye may be unable to cover the surface of amyloids that associate into larger aggregates during fibrilization (*Wetzel et al., 2018*), and its ability to bind to oligomers or small protofibrils may be limited (*Biancalana & Koide, 2010*). Furthermore, there are reports showing that ThT can modulate aggregation rates (*Carlo et al., 2015*; *Xue et al., 2017*). Finally, it is known that long exposures to light (*Wetzel et al., 2018*), elevated temperatures, and higher pH values (*Foderà et al., 2008a*; *Foderà et al., 2008b*) can cause chemical modifications to ThT molecules, thus diminishing their use in lengthy real-time observation experiments.

Another way to quantify the existence of fibrils is through measurements of solution's turbidity, mainly by light scattering (*Saiki & Akimoto, 2015*) or sample's optical density (*Roostaee, Côté & Roucou, 2009*). Large amyloid aggregates scatter incident light very effectively and their association/dissociation can be seen by changes in the intensity of scattered light. Unlike the ThT-assay, however, the intensity of scattered light does not correlate to the concentration of fibrils present in the sample in a linear way due to the inhomogeneous nature of these aggregates (*Minton, 2007*). Thus, a gradual assembly of short fibrils into long ones or the appearance of aggregate clusters may not result in a similar signal intensity pattern as a ThT assay. The one advantage it has over the previously mentioned method is that it can detect protofibrils or larger oligomers, which cannot effectively bind ThT, but are still large enough to be distinguishable from native oligomeric forms, such as dimers or tetramers (*Nettleton et al., 2000*).

A potential alternative spectroscopic method could be the recently reported phenomenon observed during examination of the fluorescence of peptide and protein aggregates using several near-UV wavelengths (*Shukla et al., 2004*; *Del Mercato et al., 2007*). When the amyloid proteins are in their native state, no unusual fluorescence emissions are seen, apart from the ones caused by aromatic amino acids. However, upon the formation of fibrils, a large fluorescence region appears at the near-UV, blue region of the excitation-emission matrix (EEM) with emissions seen up to wavelengths of 500 nm and more (*Tcherkasskaya, 2007*; *Chan et al., 2013*). These unusual emissions were termed deep-blue autofluorescence (dbAF) and have been a subject of debate regarding their possible origin and relation to amyloid fibrils (*Niyangoda et al., 2017*). A few possible hypotheses were brought up, one of which claims that the cause of dbAF are hydrogen bonds within the highly structured fibril (*Pinotsi et al., 2016*; *Del Mercato et al., 2007*). Another study claims that carbonyl groups are responsible for this phenomenon and that, to a certain extent, such fluorescence exists within non-aggregated proteins as well (*Niyangoda et al., 2017*). A possible explanation that does not involve actual fibrilization is oxidation of certain amino acids within proteins that compose the fibrils, as many fibrilization reactions involve elevated temperatures. This could lead to an appearance of optical properties in this deep-blue region (*Tikhonova et al., 2018*).

This phenomenon may prove valuable for assessing the formation or dissociation of aggregates. The reason why a dbAF assay would be a useful addition to other methods is its possible origin. If the cause for such fluorescence is hydrogen bonding in the fibrillar

structure, then its intensity should be proportional to the concentration of these bonds within a sample, thus allowing for the detection of both structures that are large enough to bind ThT and ones that are not, while at the same time retaining a linear dependence between signal strength and amyloid structure concentration. This would also allow to determine the smallest amyloid forms which may be unable to bind ThT. If the cause of this phenomenon is another factor, such as aromatic amino acid oxidation, then it should become apparent when comparing it with other amyloid formation/dissociation tracking methods.

In this work we used insulin (*Nielsen et al., 2001*) and lysozyme (*Swaminathan et al., 2011*), which are commonly used as models of amyloid formation, as well as prion protein, which is related to transmissible spongiform encephalopathies (*Torres et al., 2013*), to study the potential of deep-blue autofluorescence for monitoring amyloid fibril formation and dissociation. The data obtained using ThT, light scattering and dbAF assays were compared.

## MATERIALS & METHODS

### Fibril preparation

Human recombinant insulin (Sigma-Aldrich cat. No. 91077C) was dissolved in 100 mM sodium phosphate buffer with 100 mM NaCl (pH 2.4) to a final concentration of 0.5 mg/mL. The solution (one mL) was incubated in test tubes (Fisher cat. No. 15432545) at 60 °C for 24 h with no agitation using Ditabis MHR 23 thermoshaker. Prior to stability assays, the prepared fibrils were centrifuged for 30 min at $10,000 \times$ g and resuspended in 50 mM sodium phosphate buffer (pH 6.0) to a final volume of 1 ml. Right before further experiments the samples were sonicated for 10 min with 30 s sonication/rest intervals, using 50% sonication amplitude with a Bandelin Sonopuls sonicator (MS73 tip).

Hen egg-white lysozyme (Sigma-Aldrich cat. No. L6876) was dissolved in 50 mM sodium phosphate buffer (pH 6.0) containing 2 M guanidine hydrochloride (GuHCl) to a final concentration of 0.5 mg/ml. The solution (1 mL) with two 3 mm glass-beads (Merck, cat. No. 1040150500) added, was incubated in test tubes at 60 °C for 3 days with constant shaking at 600 rpm using the thermoshaker. Right before further experiments the samples were treated the same as the insulin fibrils.

Mouse recombinant prion protein C-terminal fragment (MoPrP89-230) was purified as described previously (*Milto, Michailova & Smirnovas, 2014*), dialyzed in Milli-Q water, lyophilized and stored at −20°C. It was dissolved in 50 mM sodium phosphate buffer (pH 6.0) containing 2 M GuHCl to a final concentration of 0.5 mg/ml. The solution (one mL) was incubated in test tubes at 60 °C for 3 days with constant shaking at 600 rpm using the thermoshaker. Right before further experiments MoPrP89-230 fibrils were resuspended the same as insulin samples and sonicated for 10 min with 30 s sonication/rest intervals, using 20% sonication amplitude with MS72 tip.

### Aggregation kinetics

Insulin, lysozyme and prion protein were dissolved as described in the fibril preparation section, and their respective preformed and sonicated aggregates (0.01%, 0.5% and 2% of total protein concentration for insulin, lysozyme and mouse prion protein respectively)

were added right before the measurement in order to reduce the influence of stochastic nucleation events (*Foderà et al., 2008a*; *Foderà et al., 2008b*) on each repeat and allow for an accurate assessment of any differences between methods.

The samples were incubated at 60 °C in a sealed 10 mm pathlength cuvette (2 mL) with low rate stirring (2 × 7 mm magnetic stirrer bar). Right-angle light scattering was measured every minute using 600 nm wavelength incident light by Varian Cary Eclipse spectrophotometer, with an excitation slit of 2.5 nm and emission slit—1.5 nm.

For dbAF measurements the samples were excited with 375 nm wavelength incident light and emission was recorded every minute at 428 nm (10 nm excitation slits were used for all measurements, 5 nm emission slits were used for insulin and lysozyme, and 10 nm emission slits –for prion protein). For this work, a dbAF fluorescence region was chosen based on the work by *Niyangoda et al. (2017)*. The excitation wavelength was high enough to not trigger any aromatic amino acid fluorescence and the emission Stoke's shift was large enough for Rayleigh or Raman scattering to only have a 10% or lower influence on total signal intensity at the end of each reaction (Fig. S1).

For the ThT assay, 10 mM ThT solution was added to each sample to a final concentration of 50 M. Samples were excited with 440 nm wavelength incident light and emission was recorded every minute at 480 nm (5 nm excitation slits were used for all measurements, 2.5 nm emission slits were used for insulin, and 5 nm emission slits –for lysozyme and prion protein cases).

## Dissociation assays

The prepared fibrils were diluted 5-fold using 50 mM sodium phosphate buffers (pH 6.0) with or without 6 M guanidine thiocyanate (GuSCN) into a range of GuSCN concentrations. Before the assay, each sample was incubated for 1 h at 25 °C.

Right-angle light scattering was measured in a micro cuvette (100 µL) using 600 nm wavelength incident light by Varian Cary Eclipse spectrophotometer, with an excitation slit of 5 nm and emission slit—2.5 nm. For each GuSCN concentration, 3 separate measurements were taken, and the emission intensities were averaged.

For dbAF measurements the samples were excited with 375 nm wavelength incident light and emission was recorded at 428 nm (10 nm excitation and 10 nm emission slits were used). For each GuSCN concentration, 3 separate measurements were taken, and the emission intensities were averaged.

For the ThT assay, 1 mM ThT solution (using higher ThT stock concentration leads to precipitates in GuSCN) was added to each sample to a final concentration of 50 M, samples (100 µL) were placed in a 96 well plate and incubated for an additional hour to allow for ThT binding at 25 °C. ThT was excited with 440 nm wavelength incident light and emission was recorded at 480 nm using Biotek Synergy H4 Hybrid Multi-Mode microplate reader. 3 separate measurements were taken, and the emission intensities were averaged.

## Atomic force microscopy (AFM)

A total of 0.5 mg/mL protein fibril solutions (non-sonicated and sonicated) were diluted 10 times. Then, 20 L of each solution were deposited onto freshly etched mica surface

and incubated for 45 s. Subsequently, samples were rinsed with 2 ml of MilliQ water and dried under gentle airflow. AFM images were recorded using Dimension Icon (Bruker) atomic force microscope operating in tapping mode and equipped with a silicon cantilever Tap300AI-G (40 N/m, Budget Sensors) with a typical tip radius of curvature of 8 nm. Images of sample topography were recorded at high-resolution (4 × 4 µm, 1,024 × 1,024 pixels). The scan rate was 0.5 Hz. Three-dimensional maps were flattened using NanoScope Analysis (Bruker) software.

## RESULTS

### Insulin aggregation and dissociation

The ThT (Fig. 1A), light scattering (Fig. 1B) and dbAF (Fig. 1C) assays show very similar aggregation kinetics with nearly identical fibrilization midpoints (18.6 ± 1.1, 18.1 ± 1.3, and 18.1 ± 1.4 min, respectively). However, curve fitting according to (*Nielsen et al., 2001*) description reveals different apparent rate constants. ThT assay suggests the slowest fibril growth rate ($k_{app} = 0.34 \pm 0.05$ min$^{-1}$), dbAF assay—slightly faster (0.41 ± 0.05 min$^{-1}$), and the light scattering assay hints towards the fastest growth rate value (0.50 ± 0.07 min$^{-1}$).

AFM revealed that sonicated insulin fibril seeds are in range of 100-300 nm in length and 3-10 nm in diameter and tend to form small clusters (Fig. S2A). The diameter distribution of elongated fibrils is similar, while the length increases to 800–2,000 nm (Fig. S2B).

Similar denaturation midpoints of insulin fibrils are determined by ThT (Fig. 1D), light scattering (Fig. 1E) and dbAF (Fig. 1F) assays (1.1, 1, and 1 M of GuSCN, respectively).

### Prion protein aggregation and dissociation

ThT aggregation curve (Fig. 2A) shows an initial gradual increase in fluorescence intensity, followed by a much more rapid increase, which reaches a plateau between 150 and 200 min. Similar to the ThT assay, light scattering (Fig. 2B) and dbAF (Fig. 2C) signal intensity increases during the initial stages of the aggregation, however, a noticeable drop in intensity before the second, more rapid increase occurs. There is also a clear difference at the end of the reaction. At the time when the plateau is expected, the light scattering and dbAF signal intensities continue to slowly rise, which makes aggregation midpoint comparison difficult due to the unclear aggregation endpoint.

AFM revealed that sonicated prion protein fibril seeds are in range of 50–400 nm in length and 3–10 nm in diameter (Fig. S3A). The diameter distribution of elongated fibrils is similar, while the length increases to 250–1,500 nm (Fig. S3B).

The stability data curves follow a similar sigmoidal tendency as in the insulin assay, however, the ThT assay (Fig. 2D) shows a slightly lower dissociation midpoint when compared to light scattering (Fig. 2E) and dbAF (Fig. 2F) assays (1.6, 1.8, and 1.7 M of GuSCN, respectively).

### Lysozyme aggregation and dissociation

The light scattering (Fig. 3B) and dbAF (Fig. 3C) assays show very similar aggregation kinetics with nearly identical fibrilization midpoints (47.7 ± 3.9, and 47.6 ± 1 min,

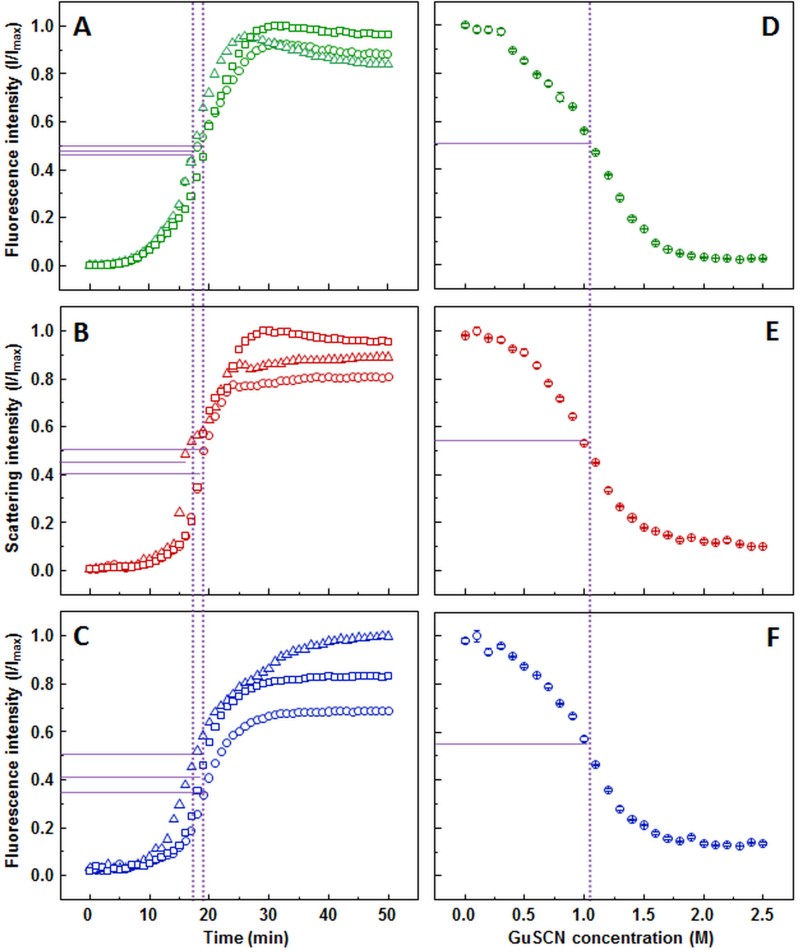

**Figure 1** **Insulin aggregation and dissociation.** Seeded aggregation of insulin with 0.01% fibrils added in the initial solution, measured by ThT (A), light scattering (B) and dbAF (C). Dissociation of insulin fibrils in GuSCN (pH 6.0), examined by ThT (D), light scattering (E) and dbAF assays (F). Signal intensity is displayed as a fraction of maximum intensity of all three repeats. Solid purple lines show the halfway between its respective minimum and maximum value, dotted purple lines represent the time or denaturant concentration at which ThT fluorescence intensity reaches this value.

respectively), while the one determined by ThT (Fig. 3A) assay is lower (37.3 ± 3.3 min). The ThT and light scattering data points are slightly more dispersed than in the dbAF assay. The shape of the kinetic curves is different at the early stages of aggregation, with ThT assay suggesting a more rapid aggregation without any lag time, when compared to the other two methods.

AFM revealed that sonicated lysozyme fibril seeds are in range of 150-300 nm in length and up to 15 nm in diameter (Fig. S4A). It is problematic to measure the diameter of a single fibril due to intense clustering. The elongated fibrils are in range of 3–15 nm in diameter, and 400–2,000 nm in length (Fig. S4B).

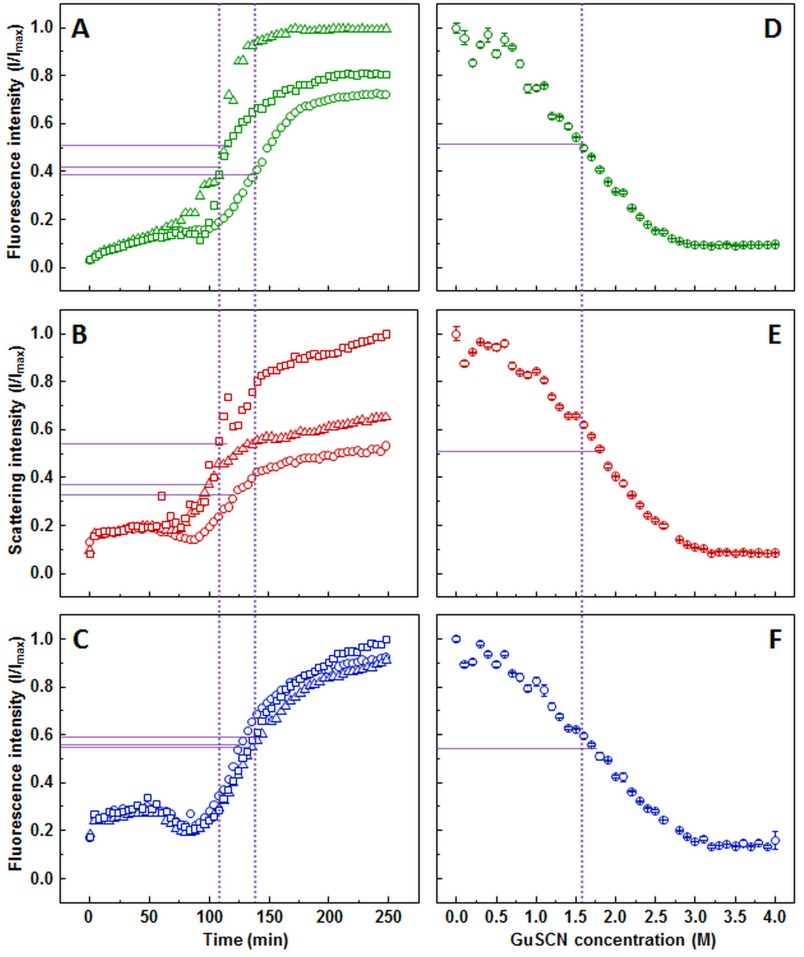

**Figure 2   Prion protein aggregation and dissociation.** Seeded aggregation of mouse prion protein with 2% fibrils added in the initial solution, measured by ThT (A), light scattering (B) and dbAF (C). Dissociation of mouse prion protein fibrils in GuSCN (pH 6.0), examined by ThT (D), light scattering (E) and dbAF assays (F). Signal intensity is displayed as a fraction of maximum intensity of all three repeats. Solid purple lines show the halfway between its respective minimum and maximum value, dotted purple lines represent the time or denaturant concentration at which ThT fluorescence intensity reaches this value.

Lysozyme fibril dissociation curve observed by ThT (Fig. 3D) has a scattered signal at lower denaturant concentrations compared to the case of insulin (Fig. 1D), however the sigmoidal shape of the curve allows determination of the dissociation midpoint (2.2 M GuSCN). The light scattering (Fig. 3E) and dbAF (Fig. 3F) assay signals are gradually decreasing starting from the lowest GuSCN concentration, followed by a steeper drop between 2.5 and 3.5 M of GuSCN and a plateau. Such results make it difficult to discern the possible fibril dissociation midpoint, whether it is at the half intensity point of the entire curve or the steeper drop part. In either case, the stability midpoint is considerably lower in the ThT assay.

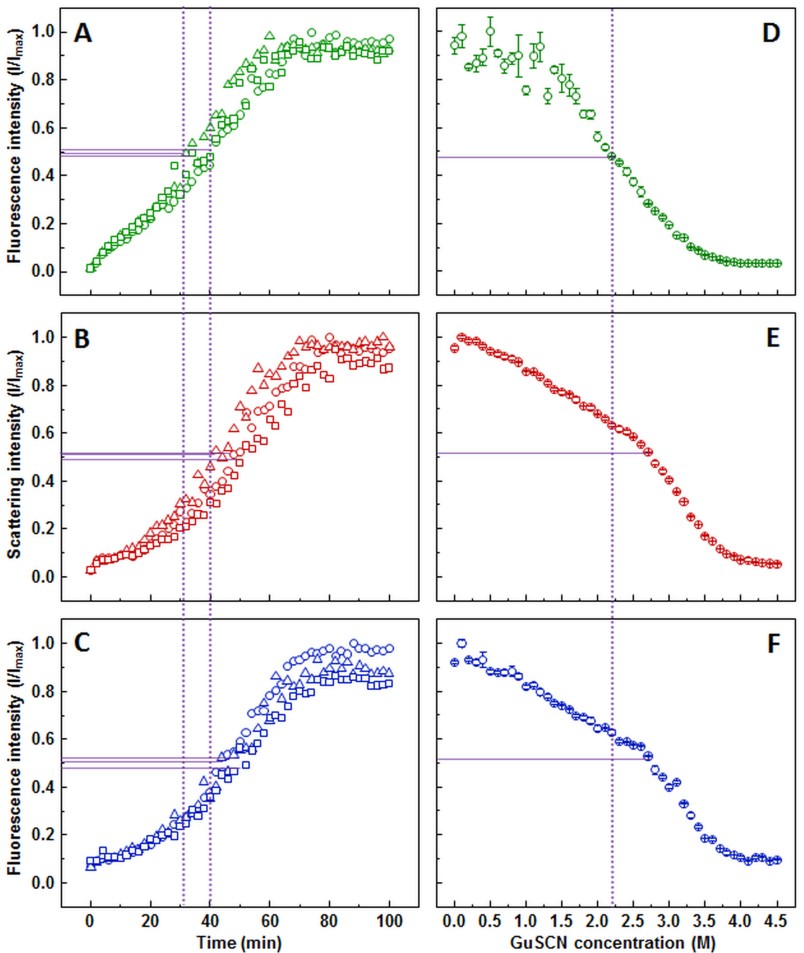

**Figure 3** **Lysozyme aggregation and dissociation.** Seeded aggregation of lysozyme with 0.5% fibrils added in the initial solution, measured by ThT (A), light scattering (B) and dbAF (C). Dissociation of lysozyme fibrils in GuSCN (pH 6.0), examined by ThT (D), light scattering (E) and dbAF assays (F). Signal intensity is displayed as a fraction of maximum intensity of all three repeats. Solid purple lines show the halfway between its respective minimum and maximum value, dotted purple lines represent the time or denaturant concentration at which ThT fluorescence intensity reaches this value.

## DISCUSSION

### Comparison of aggregation kinetics

Kinetic data on insulin aggregation suggests that all three used methods are equally viable for determination of fibrilization midpoint; however, one must be careful comparing apparent rate constants as these parameters may be differently interpreted by certain methods. Looking into the origins of each method one can think that ThT fluorescence rises with increased surface area of aggregates, and light scattering signal grows with increased size of aggregates, while dbAF may rise with the number of specific bonds. If this is true, dbAF would be the most reliable method to follow insulin aggregation kinetics.

Out of the three methods used to follow lysozyme aggregation kinetics, the one that gives different results from the rest is the ThT assay. The ThT curve has a rather exponential shape, which can be interpreted as an elongation-driven process, while both light scattering and dbAF curves are sigmoidal, suggesting a bigger role of nucleation. This can lead to a simple conclusion that lysozyme aggregation was affected by the presence of ThT. Non-linear dependence of light scattering on the size of aggregates could also explain the differences with the ThT assay. If bonds, involved in 3-dimentional association of fibrils contribute to dbAF, it would also support such an explanation.

In the case of prion protein, shapes of kinetic curves are puzzling. As each sample has 2% preformed fibrils added, we would expect to see a continuous growth from the very beginning and the plateau at the end. ThT data meets the expectations, and the rapid growth phase between 100 and 150 min suggests a strong role of nucleation. Light scattering and dbAF curves display some important differences: the small rapid increase within the first few minutes, decrease right before the main growth phase, and inability to reach the plateau. The latter suggests slow association of aggregates, after the fibril elongation finishes, leading to the increased size of fibrils and more intermolecular bonds. The other two differences are harder to explain. Monomer binding to the preformed fibrils can be a possible cause for the increased light scattering and dbAF. If bound monomers form nuclei within a certain time and detach from the fibril surface, it could serve as an explanation for the decreased signal.

## Comparison of fibril dissociation measurements

When examining insulin dissociation each assay appears to be a viable option. The only concern is a slight drift of the light scattering signal after the plateau should be reached.

Slight variation in dissociation midpoint of prion protein fibrils was observed among used methods with lowest value suggested by the ThT assay. Additional experiments with light scattering and dbAF proved that the difference is not caused by longer incubation, used in ThT protocol (Fig. S5).

The lysozyme dissociation curves are more contrasting between methods. ThT curve has a regular sigmoid shape, similar as in case of other proteins. The initial decrease in both scattering and dbAF signal intensity may be due to fibril cluster (Fig. S4) dispersion with increasing GuSCN concentration, as it is unlikely that fibrils prepared in 2 M GuHCl would be unstable at very low GuSCN concentrations. The steeper drop in intensity at higher denaturant values is most likely the actual point at which the fibrils begin to dissociate into monomers. Such shapes of the signal curves make it difficult to distinguish the actual stability midpoint. Despite this, however, both the midpoint of the entire curve and the midpoint of the steep drop suggest a considerably higher denaturant concentration compared to the ThT assay. Such a difference means that some of the tested methods may give improper results. To resolve this issue, we have employed an alternative method to follow fibril dissociation (Fig. S6). Obtained data is comparable with the steep drop in light scattering and dbAF curves, which suggests that ThT data is misleading. It seems that higher denaturant concentrations affect ThT binding to amyloid fibrils or quenches its fluorescence, which leads to a lower apparent dissociation midpoint when measured by ThT assay.

### Is dbAF related to aggregate size?

While the ThT and light scattering assay results are quite different in some cases, it is impossible to not notice the similarity between light scattering and dbAF data, especially in the lysozyme dissociation assay and prion protein aggregation kinetics. The large wavelength difference between incident and emission light in the dbAF assay reduces the effects of any light scattering to 10% or less of total signal intensity (Fig. S1), which makes it even stranger that there is such a high correlation between the two measurements. The dissociation curves remain nearly identical even when a baseline correction is applied to negate any light scattering effects (Fig. S7). Perhaps it is possible that the dbAF phenomenon is more closely related to the size of aggregates, as we observe not only an increased length of fibrils after the seeded growth reaction occurs, but also fibril clumping into clusters, especially visible with lysozyme samples (Fig. S4).

Similar to a previously mentioned work by (*Niyangoda et al., 2017*) where dbAF emissions were seen for non-aggregated proteins, dbAF emission intensity of dissociated protein samples does not reach the background noise level in the scanned range. The emission maxima position retains a certain intensity at high GuSCN concentrations, even when a dbAF intensity plateau is reached (which would indicate complete denaturation), suggesting that these emissions may be enhanced not by amyloid structure formation, but by the increasing size of protein aggregate assemblies.

We also have to make a point that there have been reports showing protein crystals possessing dbAF (*Shukla et al., 2004*), which does not rule out the possibility of non-amyloid structures possessing this phenomenon. This means there may not be a noticeable distinction between an amyloid fibril and an amorphous aggregate or protein crystal. It was also shown that during unseeded protein aggregation measurements, dbAF intensity begins to increase at the early stages of the reaction, well before any ThT signal or sample turbidity changes are observed (*Tikhonova et al., 2018*). This could be caused by the formation of oligomers or small protofibrils, an event which would be less likely in seeded aggregation used in this work.

## CONCLUSIONS

Our results show that dbAF can be used to follow aggregation and dissociation of different proteins and has similar limitations as light scattering. The case of lysozyme dissociation has demonstrated possible divergence of the same process followed by different methods, advocating for parallel usage of at least two alternative methods when possible.

### Funding

This research was funded by the grant no. TAP LLT-1/2017 from the Research Council of Lithuania. The funders had no role in study design, data collection and analysis, decision to publish, or preparation of the manuscript.

## Grant Disclosures

The following grant information was disclosed by the authors:
Research Council of Lithuania: TAP LLT-1/2017.

## Competing Interests

The authors declare there are no competing interests.

## Author Contributions

- Mantas Ziaunys conceived and designed the experiments, performed the experiments, analyzed the data, prepared figures and/or tables, authored or reviewed drafts of the paper, approved the final draft.
- Tomas Sneideris performed the experiments, analyzed the data, authored or reviewed drafts of the paper, approved the final draft.
- Vytautas Smirnovas conceived and designed the experiments, analyzed the data, contributed reagents/materials/analysis tools, authored or reviewed drafts of the paper, approved the final draft.

## Data Availability

The raw data are available in the Supplemental Files.

## Supplemental Information

Supplemental information for this article can be found online at http://dx.doi.org/10.7717/peerj.7554#supplemental-information.

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
