# Peer review of "Exploring the potential of deep-blue autofluorescence for monitoring amyloid fibril formation and dissociation"

_PeerJ, doi:10.7717/peerj.7554_

## Round 0.1 · original submission · Major Revisions

Please address all critiques of both reviewers, paying careful attention to the comments of the reviewer #1, and amend your manuscript accordingly.

Reviewer 1 ·

Basic reporting

no comment

Experimental design

The deep-blue autofluorescence in amyloid fibrils even when formed by proteins without aromatic amino acids is an intriguing observation. Yet the origin of this signal and, therefore, its potential to provide additional information about the amyloid assembly process remain controversial. Additional experiments to elucidate both the origin and utility of this signal are certainly welcome. The authors try to explore the utility of dbAF for fibril growth kinetics by combining deep-blue autofluorescence measurements with light scattering and ThT measurements of the kinetics of seeded growth, and GuSCN-induced dissociation, of aggregates formed by insulin, lysozyme and human prion protein. One of the intrinsic weaknesses of this paper is that it remains mostly descriptive without providing additional insights into the origin of the dbAF signal. The authors do suggest that aggregate size might play an important role – but don’t provide any independent evidence to support this hypothesis.

Important Concerns/ Requested Changes

1) Perhaps the biggest omission is the lack of characterization of the morphologies and size distributions of (a) their amyloid seeds under their conditions used for seeded growth or (b) of the aggregate morphologies at the end of seeded growth. For one, the resulting aggregate populations might actually be quite heterogeneous, which would raise the question which ones are responsible for the observed changes in any of the three readouts. In addition, the growth conditions (pH 6) are quite unusual in particular for insulin and lysozyme. This might further complicate the interpretation if the aggregates are complex mixtures of various distinct populations. Even if fairly uniform, the overall structure/morphology/size of the aggregates prior to and following seeded growth might play an important role in the observed dbAF kinetics.

2) The authors indicate that the large (50 nm) Stokes shift should preclude a leak from light scattering into their dbAF fluorescence channel. Given that the authors work with probably large aggregates formed during seeded growth and that scattering intensities from short-wavelength excitation can be orders of magnitude larger (even far from the excitation source) than any fluorescence signal, the authors need to confirm this assertion directly. This is particularly important since the light scattering and dbAF signals show noticeable correlations in their time course for all three proteins.
One way to check this would be to measure the actual spectrum of their dbAF emission. If that spectrum rides on a large shoulder that increases towards the short-wavelength cut-off ( i.e. towards the excitation wavelength ) this would indicate that scattered excitation light is leaking into their dbAF emission channel – despite the significant Stokes shift.

Minor Concerns/ Suggestions

3) Instead of displaying their signals on an arbitrary scale (0-100) without a clear reference point, the authors should represent their signals as fractional change ( i.e. S / S(0) or [S-S(0)] / S(0)). This permits a comparison of the relative changes of any of these three signals during growth and enables readers to compare results obtained with different instruments against those of the authors.

4) The authors provide no clear motivation why they chose to follow seeded growth vs. monitoring the nucleated polymerization of their seeds in the first place. This deserves some comment.

Validity of the findings

some of the comments above relate to the question whether the reported data are sufficient to support the author's conclusion.

Reviewer 2 ·

Basic reporting

Article conforms to the basic reporting guidelines of PeerJ

Experimental design

The research is within the scope of the journal. Research question in relevant for study protein aggregation.

Validity of the findings

Data is provided and is sound and conclusions are stated well.

Additional comments

In the figures1,2 and 3, for the aggregation studies, each of A, B & C have three sets of data points. It may be prudent to plot each of them in a different color to help the reader identify that there are three different kinetic experiments.

Alternatively one could plot them as mean+/SD too.

It may also be useful for the reader to see a quantitative comparison between the three different methods and may be achieved by fitting the data to a kinetic model.

---

## Round 0.2 · Minor Revisions

Please address the remaining concerns of the reviewer #1.

Reviewer 1 ·

Basic reporting

see general comments

Experimental design

see general comments

Validity of the findings

see general comments

Additional comments

I have reviewed the revisions of the original manuscript. There are still some short-comings - but I would like to leave it to the authors to address those residual issues at their discretion. I am particularly pleased with the added AFM images which give a nice overview of the changes in aggregate morphology. I would further agree that these images do show predominately fibrillar species.

Residual issues/concerns
1) Concern about contributions from light scattering to dbAF. The authors have performed the experiments I suggested - but I do disagree with their assessment. Looking at the panels of their supplemental Figures S1D-F, it is obvious that their dbAF signal (black) is indeed riding on a LARGE shoulder towards their excitation wavelengths. It is this exact shoulder in their fluorescence signal that they need to account for - not the relatively minor contributions at the beginning of their experiments (initial condition?). Their (uncontaminated) fluorescence signal should have a nearly flat baseline on either side of the dbAF peak (see the shape of the dbAF peak in S1A). To correct this properly, they should use a two-peak gaussian (or Lorentzian) fit to their fluorescence trace - with one peak centered at the dbAF peak and the other centered at or near the excitation wavelength (which represents the shoulder in their spectra). My rough estimate is that the corresponding contribution to their dbAF peak in eg. panel S1F is closer to 50%. This might significantly affect their interpretation of the dbAF vs. light scattering kinetics.

2) I could not find any figure captions to the supplemental figures. The authors should make sure those get included in the final manuscript.

3) Seeded vs. unseeded kinetics: The investigation of exclusively seeded growth kinetics does raise some questions as to whether dbAF is a good readout of amyloid kinetics in the absence of seeding. The authors might want to elude to this issue in their discussion. The reason I add this cautionary note this is that, in our hands, dbAF did NOT replicated unseeded fibril kinetics with either ThT or light scattering.

Reviewer 2 ·

Basic reporting

No comment

Experimental design

No comment

Validity of the findings

No comment

Additional comments

I am satisfied with the response of the authors.

---

## Round 0.3 · accepted · Accept

Thank you for addressing the remaining issues of the reviewers and for the final amendments of the manuscript. Your revised manuscript is acceptable now.